



# Derivation and compilation of lower atmospheric properties relating to temperature, wind, stability, moisture, and surface radiation budget over the central Arctic sea ice during MOSAiC

Gina C. Jozef[1,2,3], Robert Klingel[4], John J. Cassano[1,2,3], Björn Maronga[4,5], Gijs de Boer[2,6,7],
Sandro Dahlke[8], Christopher J. Cox[6]

[1]Dept. of Atmospheric and Oceanic Sciences, University of Colorado Boulder, Boulder, CO, USA
[2]Cooperative Institute for Research in Environmental Sciences, University of Colorado Boulder, Boulder, CO, USA
[3]National Snow and Ice Data Center, University of Colorado Boulder, Boulder, CO, USA
[4]Leibniz University Hannover, Institute of Meteorology and Climatology, Hannover, Germany
[5]Geophysical Institute, University of Bergen, Bergen, Norway
[6]NOAA Physical Sciences Laboratory, Boulder, CO, USA
[7]Integrated Remote and In Situ Sensing, University of Colorado Boulder, Boulder, CO, USA
[8]Alfred Wegener Institute Helmholtz Centre for Polar and Marine Research, Potsdam, Germany

*Correspondence to:* Gina Jozef (gina.jozef@colorado.edu)

**Abstract.** Atmospheric measurements taken over the span of an entire year between October 2019 and September 2020 during the icebreaker-based Multidisciplinary drifting Observatory for the Study of Arctic Climate (MOSAiC) expedition provide insight into processes acting in the Arctic atmosphere. Through the merging of disparate, yet complementary in situ observations, we can derive information about these thermodynamic and kinematic processes with great detail. This paper describes methods used to create a lower atmospheric properties dataset containing information on several key features relating to the central Arctic atmospheric boundary layer, including properties of temperature inversions, low-level jets, near-surface meteorological conditions, cloud cover, and the surface radiation budget. The lower atmospheric properties dataset was developed using observations from radiosondes launched at least four times per day, a 10 m meteorological tower and radiation station deployed on the sea ice near the Research Vessel *Polarstern*, and a ceilometer located on the deck of the *Polarstern*. This lower atmospheric properties dataset, which can be found at *insert DOI when published*, contains metrics which fall into the overarching categories of temperature, wind, stability, clouds, and radiation at the time of each radiosonde launch. The purpose of the lower atmospheric properties dataset is to provide a consistent description of general atmospheric boundary layer conditions throughout the MOSAiC year which can aid in research applications with the overall goal of gaining a greater understanding of the atmospheric processes governing the central Arctic and how they may contribute to future climate change.

## 1 Introduction

The Arctic is warming at a rate at least twice that of the rest of the planet (Overland et al., 2019), a phenomenon called Arctic amplification (Serreze and Francis, 2006; Serreze and Barry, 2011), which has significant consequences both for the Arctic and across the globe (Serreze and Barry, 2011; Cohen et al., 2014; Coumou et al., 2018). The sea ice albedo feedback is a recognized and well-studied contributor to a disproportionately warming Arctic (Winton, 2006; Jenkins and Dai, 2021), leading directly to increased outgoing longwave radiation and turbulent heat fluxes from



newly open ocean (Dai et al., 2019). However, processes in the lower atmosphere, which can indirectly contribute to Arctic warming and the way that warming is distributed, are poorly understood (Tjernström et al., 2012) and less frequently studied. This lack of understanding contributes to inaccuracies in the representation of present-day sea ice (Stroeve et al., 2012) and uncertainties in the state of the future Arctic climate in climate models (Hodson et al., 2012; Karlsson and Svensson, 2013; Cai et al., 2021). Determining the predominant thermodynamic structures and kinematic processes occurring in the Arctic lower atmosphere, and how these relate to cloud characteristics and radiative transfer, may help to constrain some of these uncertainties.

Insight into prevalent Arctic atmospheric processes can be gained by analysis of data collected during the MOSAiC (Multidisciplinary drifting Observatory for the Study of Arctic Climate) expedition (Shupe et al., 2020). MOSAiC was a year-long expedition that took place from October 2019 to September 2020 in which the Research Vessel *Polarstern* (Alfred-Wegener-Institut Helmholtz-Zentrum für Polar- und Meeresforschung, 2017) was frozen into the central Arctic Ocean sea ice pack and allowed to passively drift across the central Arctic for an entire year. During MOSAiC, a variety of instruments were deployed from the *Polarstern*, the section of sea ice approximately next to the *Polarstern* (hereafter called the MOSAiC floe), and at distances up to 40 km from the *Polarstern* (called the distributed network; Krumpen and Sokolov, 2020). A core goal of MOSAiC was to study key processes occurring in the atmosphere (Shupe et al., 2022), sea ice (Nicolaus et al., 2022), and ocean (Rabe et al., 2022) to understand Arctic climate change. Between October 2019 and mid-May 2020, the *Polarstern* drifted with the original MOSAiC ice floe. In mid-May, the *Polarstern* left the MOSAiC floe to conduct an exchange of people and equipment in Svalbard, and returned to the original MOSAiC floe in mid-June, where it remained until the end of July. At this point, the original MOSAiC floe disintegrated, so the *Polarstern* relocated to a newly identified ice floe near the North Pole, where it remained from late August through late September (Shupe et al., 2022).

The purpose of this paper is to summarize the methods used to develop a lower atmospheric properties dataset (*insert DOI when published*; Jozef et al., 2023) containing important information on several key atmospheric features, including the atmospheric boundary layer (ABL), temperature inversions (TIs), low-level jets (LLJs), near-surface meteorological state, cloud cover, and surface radiation budget over the span of an entire year in the Arctic. This lower atmospheric properties dataset was developed by identifying features in all balloon-borne radiosondes, which were launched several times per day over the span of the MOSAiC year from the deck of the *Polarstern,* and supplemented by near-surface atmospheric data from a 10 m meteorological tower and surface radiation data from the radiation station located on the MOSAiC floe, as well as information on cloud cover from a ceilometer located on the deck of the *Polarstern*.

This paper does not delve into the physical significance of these observations. Rather, the goal is simply to explain the instrumentation (Sect. 2) and methods (Sect. 3) used to develop the accompanying lower atmospheric properties dataset, in hopes that the dataset will be useful to a wide variety of other projects.



## 2 Instrumentation

### 2.1 Radiosondes

The primary platform used to develop the lower atmospheric properties dataset is the radiosonde. Throughout MOSAiC, radiosondes were launched from the stern deck of the *Polarstern* at least four times per day (every 6 hours) for the entire year. These launches were typically conducted at 05:00, 11:00, 17:00, and 23:00 UTC (Maturilli et al., 2021). During events of particular interest, such as a warm air intrusion event, or time spent sailing across the sea ice edge, radiosondes were launched up to 8 times per day (every 3 hours). Figure 1 shows the locations of each radiosonde launch throughout the expedition.

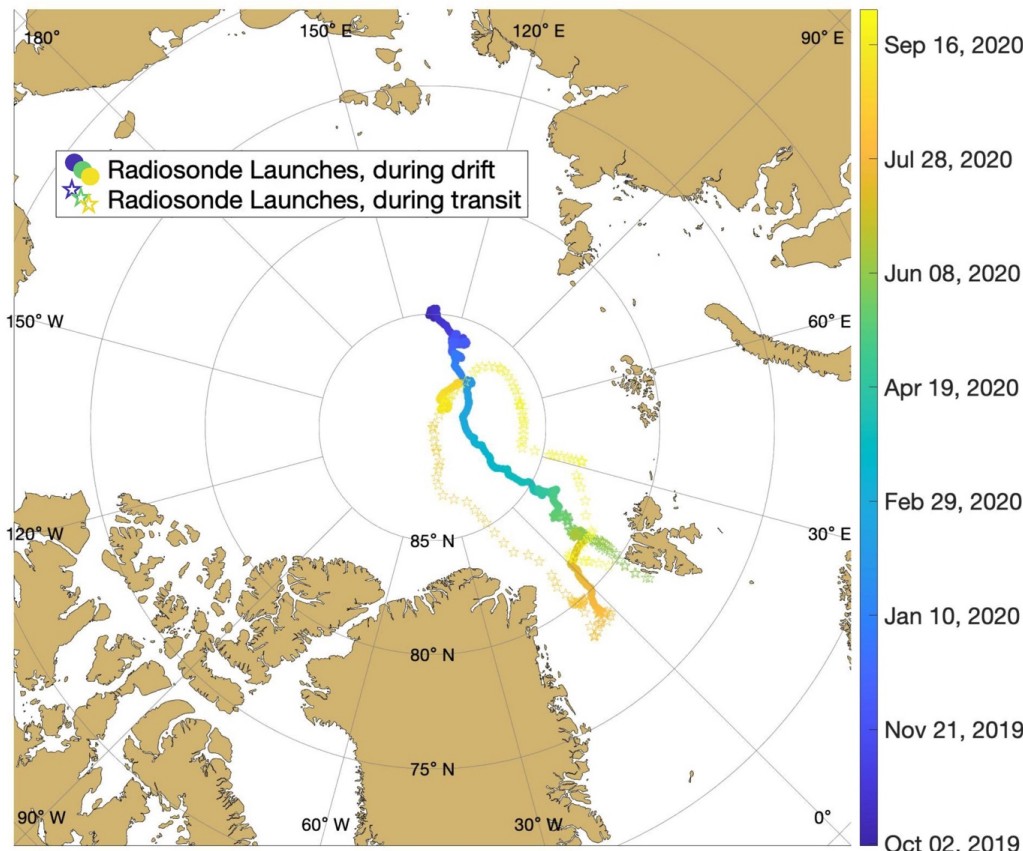

**Figure 1.** Map of the central Arctic showing the location of each radiosonde launch, color coded by date. Circular symbols indicate when the *Polarstern* was passively drifting, and star symbols indicate when the *Polarstern* was travelling under its own power.

The balloon-borne Vaisala RS41 radiosondes used during MOSAiC measured temperature, pressure, relative humidity, and wind between the helicopter deck of the *Polarstern* (~12 m above the ice and depicted in Fig. 3 of



Shupe et al. 2022) and about 30 km altitude (Maturilli et al., 2021). We use the level 2 radiosonde product (Maturilli et al., 2021) for the lower atmospheric properties dataset, as the level 2 radiosonde data are found to be more reliable in the lower troposphere than the level 3 radiosonde data (Maturilli et al., 2022). For the purpose of the lower atmospheric properties dataset, we only use measurements up to 5 km, as this is roughly the upper limit of the lower troposphere (Silva and Schlosser, 2021) and we are interested only in lower atmospheric features. Radiosonde

measurements were recorded with a frequency of 1 Hz with a typical ascent rate of 5 m s$^{-1}$, resulting in measurements approximately every 5 m throughout the ascent. Information about instrumentation uncertainty can be found in Table 2. Radiosonde measurements were used to identify and characterize several key features of the lower atmosphere including ABL depth and stability, and characteristics of TIs and LLJs. While the radiosondes profile is not an instantaneous snapshot of the atmosphere (it takes the radiosonde ~20 minutes to reach 5 km), the measurements can

still provide a reasonable representation of the atmospheric state at the time of radiosonde launch, especially near the surface. Thus, all additional variables that were not derived directly from the radiosonde measurements are provided at the time of each radiosonde launch, presented as an average of values within approximately 5 minutes before and after launch (averaging interval is explained further throughout text).

Prior to processing the radiosonde data for integration into the lower atmospheric properties dataset, radiosonde

measurements were corrected to account for the local "heat island" resulting from the presence of the *Polarstern.* This local source of heat resulted in the frequent occurrence of elevated temperatures near the launch point, resulting in inconsistencies in the observed temperatures in the lowermost part of the atmosphere. This effect was found to frequently influence radiosonde measurements up to 23 m above sea level (11 m above the *Polarstern*'s helicopter deck) and in some cases it was observed to extend even higher. This phenomenon can be recognized by a temperature

structure indicative of a convective layer below 23 m. We know from previous literature that convective ABLs are rare in the central Arctic (Tjernström et al., 2004; Brooks et al., 2017), so it is unlikely that nearly all radiosondes would exhibit thermodynamic properties associated with convection near the surface. Therefore, this part of the profile is understood to be an artifact of the contamination, and thus unreliable. To mitigate for this influence, all radiosonde measurements below 23 m were excluded. This helps in also removing faulty wind measurements that occur as a result

of flow distortion around the ship (Berry et al., 2001), and the radiosonde motion induced by the initial unraveling of the string that connects the radiosonde to the balloon.

If anomalously warm temperature measurements appeared to extend above 23 m (identified by continued presence of a convective atmosphere), then the lowest radiosonde measurements were visually compared to measurements from the 10 m meteorological tower to identify where temperature values were anomalously warm above 23 m. This was

identifiable when the tower measurements interpolated upward, given their observed slope, did not match up with the radiosonde measurement at 23 m. The first credible value of the radiosonde measurements is found when the tower measurements interpolated upward would line up with the observed radiosonde measurement, or in the case of a temperature offset between the tower and radiosonde, would have the same slope. Data at the altitudes below this first credible value were removed.



An additional disruption of the radiosonde measurements sometimes occurred as a result of the passage of the balloon through the ship's exhaust plume. When it was unambiguous that the radiosonde passed through the ship's plume (evident by a sharp increase and subsequent decrease in temperature, typically by ~0.5-1 °C over a vertical distance of ~10-30 m, identified visually), these values were replaced by values resulting from interpolation between the closest credible values above and below the anomalous measurements, which are identified as the last point just before the

increase and the first point just after the decrease in temperature values, to acquire a continuous profile of reliable temperatures.

## 2.2 Meteorological tower

Near-surface data from the 10 m meteorological tower (hereafter called the met tower) is included to provide the near-surface context at the time when features identified in the radiosondes occurred, since radiosonde measurements do

not extend to the surface. The met tower was located on the sea ice at a site called Met City (Shupe et al., 2022), which varied between 300 to 600 m from the *Polarstern* (Cox et al., submitted; Cox et al., 2023) throughout the campaign. The met tower measurements included in the lower atmospheric properties dataset were recorded at approximately 2 m and 10 m above the surface of the sea ice (the true altitudes for each variable are shown in Table 1, where the given ranges account for varying snow depths). The values of variables from the met tower included in the lower atmospheric

properties dataset were determined using the 1 minute met tower data (these data are reported as the average of the observations between the minute reported, and the following minute, e.g., data at 12:30 UTC is an average of observations between 12:30 and 12:31 UTC), averaged between 5 minutes before and 5 minutes after the time of launch, to avoid the potential for small fluctuations in the measurements at time of radiosonde launch to misrepresent the true state of the atmosphere. This is carried out by determining the met tower time stamp closest in time to the

radiosonde launch, and averaging between 5 minutes before to 5 minutes after this time. For example, for a radiosonde launch time of 11:45:15 UTC, the corresponding met tower data is averaged between 11:40 and 11:50 UTC; as the data for 11:50 is the mean between 11:50 and 11:51 UTC, this results in data averaged over an 11 minute period.

Temperature and relative humidity at the 10 m level were measured using a Vaisala HMT330, and at the 2 m level using a Vaisala PTU300; atmospheric pressure was also observed at the 2 m sensor. Wind speed and direction were

measured using a Metek uSonic-Cage MP sonic anemometer. Pressure at the 10 m level was approximated using the hypsometric equation (Stull, 1988). Information about instrumentation uncertainty can be found in Table 2.



**Table 1:** True altitudes for met tower variables. Ranges in parentheses reflect the varying snow depth.

| Met Tower Variable | True "2m" height | True "10m" height |
|---|---|---|
| Temperature | 1.75 m (1.1 – 2.2 m) | 9.44 m (8.7 – 9.8 m) |
| Relative humidity | 1.46 m (0.8 – 1.9 m) | 9.15 m (8.5 – 9.6 m) |
| Pressure | 1.54 m (1 – 2.1 m) | NA |
| Wind | 2.62 m (2 – 3.1 m) | 10.34 m (9.9 – 1.1 m) |

In addition to providing metrics only recorded by the met tower, we also include some metrics calculated using data from both the met tower and radiosonde, specifically bulk Richardson number and the change in virtual potential temperature calculated between 2 m from the met tower and the top of the ABL from the radiosondes (see Sect. 3.3). To improve the validity of such integrated quantities, work is in progress to interpolate between the tower and radiosonde measurements to create a continuous profile from the ground, which removes anomalous measurements

in the radiosonde profiles resulting from the *Polarstern*'s heat island and exhaust plume effects.

     While the radiosondes were launched at least four times per day throughout the entire MOSAiC year, met tower measurements were continuous when active; however, the met tower was not always active. This is because the met tower was located on the sea ice and needed constant power to run. Therefore, during transit periods, or times when power to the met tower was cut, we do not have these near-surface measurements. The primary times in which we do

not have met tower data are before 15 October 2019 (beginning of experiment), between 10 May and 7 June 2020 (*Polarstern* transit), between 29 July and 25 August 2020 (*Polarstern* transit), and after 18 September 2020 (end of experiment). Radiosonde and ceilometer measurements (Sect. 2.3) during these periods are relative to the position of the *Polarstern*, not to the position of the MOSAiC floe. Between the *Polarstern* transit events, the met tower was installed in different locations (varying iterations of Met City) on the ice (three in total; the first two on the original

ice floe, and the third on the newly-identified ice floe farther north), but each was always less than 600 m from the *Polarstern*.

### 2.3 Ceilometer

     Information on cloud characteristics provided in the lower atmospheric properties dataset comes from the Vaisala Ceilometer CL31 (ARM user facility, 2019) located on the P-deck of the *Polarstern* (depicted in Fig. 3 of Shupe et

al. 2022), deployed as part of the Department of Energy (DOE) Atmospheric Radiation Measurement (ARM) mobile facility suite (Shupe et al. 2021). The ceilometer measures atmospheric backscatter and cloud base height, which allows us to determine the altitude and presence of clouds during the time of radiosonde launch. The ceilometer measurements were recorded with a laser pulse rate of 10 kHz and averaged over 16 s. Information about instrumentation uncertainty can be found in Table 2. The utilized variables were determined using data averaged



between approximately 5 minutes before and 5 minutes after the time of launch, following the same time interval format as the met tower data, for which an example was given in Sect. 2.2. The averaging intervals vary due to the 16 s source data, but are kept as small as possible while ensuring the aforementioned temporal spans. Intervals with less than 50% data coverage are excluded from the follow-up calculations and marked as missing. Note that the altitude of the P-deck was approximately 20 m above sea level, which could occasionally be above the presence of fog or

blowing snow.

### 2.4 Radiation station

Information on surface radiation provided in the lower atmospheric properties dataset comes from the DOE ARM radiation station located on the MOSAiC floe adjacent to the met tower at Met City (Shupe et al., 2022). This radiation station was outfitted with Eppley Precision Infrared Radiometers for measuring downwelling and upwelling

broadband longwave radiation, and Eppley Standard Precision Pyranometers for measuring downwelling and upwelling broadband shortwave radiation (Cox et al., submitted). Information about instrumentation uncertainty can be found in Table 2. Variables from the radiation station provided at the time of radiosonde launch were determined using 1 minute data (these 1 minute data are determined in the same manner as the met tower data), averaged between 5 minutes before and 5 minutes after the time of launch, following the same time interval format as the met tower

data, for which an example was given in Sect. 2.2. Prior to averaging, radiation measurements with values outside of a reasonable range (such as large values for shortwave radiation during polar night, or negative values for any of the radiation components, explained further in Sect. 3.5) were excluded. During the times listed above in which the met tower was not taking measurements, we also do not have radiation station measurements.



**Table 2:** Variable uncertainty in the instrumentation used to derive the lower atmospheric properties dataset.

| Platform | Variable | Instrumentation | Uncertainty |
|---|---|---|---|
| **Radiosonde** | Pressure | Vaisala RS41-SGP | 1.0 hPa (> 100 hPa), 0.6 hPa (< 100 hPa) |
| | Temperature | | 0.3 °C (< 16 km) 0.4 °C (> 16 km) |
| | Relative humidity | | 4 % |
| | Wind speed | | 0.15 m s$^{-1}$ |
| | Wind direction | | 2 ° |
| **Met Tower** | 2 m temperature | Vaisala HMT337 | 0.3 – 0.4 °C |
| | 10 m temperature | | 0.3 – 0.4 °C |
| | 10 m relative humidity | | 1.6 – 1.8 % |
| | 2 m relative humidity | Vaisala PTU307 | 1.6 – 1.8 % |
| | 2 m pressure | | 0.15 hPa |
| | 2 and 10 m wind speed | Metek uSonic-Cage MP sonic anemometer | 0.3 m s$^{-1}$ |
| | 2 and 10 m wind direction | | 2 ° |
| **Ceilometer** | Cloud base height | Vaisala CL31 | 5 m |
| **Radiation station** | Longwave radiation | Eppley Precision Infrared Radiometer | 2.6 W m$^{-2}$ (downwelling) 1 W m$^{-2}$ (upwelling) |
| | Shortwave radiation | Eppley Standard Precision Pyranometer | 4.5 W m$^{-2}$ |

### 3 Variables included in the lower atmospheric properties dataset

#### 3.1 Temperature

The temperature-related variables provided in the lower atmospheric properties dataset include temperature inversion
characteristics as well as temperature (and pressure) from the met tower and at the ABL top (derivation of ABL height
is discussed in Sect. 3.3).

To identify a TI layer, we refer to a profile of the temperature gradient (dT/dz) for each case. dT/dz is calculated across
30 m intervals in steps of 5 m and attributed to the center altitude of Δz (i.e., 23-53 m, 28-58 m, 33-63 m and so on,
resulting in a dT/dz profile with values at 38 m, 43 m, 48 m AGL, and so on) between the bottom of the radiosonde
profile and 5 km. We then determine the presence of a TI layer by identifying where dT/dz exceeds a threshold of
0.65 °C (100 m)$^{-1}$. Previous work by Kahl (1990) and Gilson et al. (2018) use a threshold of 0 °C (100 m)$^{-1}$, however,
we find that a threshold of 0.65 °C (100 m)$^{-1}$ is better suited for the fine scale vertical resolution of the radiosonde
data. In 28% of cases, using a threshold of 0.65 instead of 0 °C (100 m)$^{-1}$ does not make a difference in what is
identified as a TI layer. However, in most instances using the higher threshold is critical. If we use a threshold of 0 °C
(100 m)$^{-1}$ and identify anywhere where dT/dz exceeds this threshold as a TI (Kahl, 1990), then we can incorrectly
identify a nearly isothermal layer as a TI. Using a threshold of 0.65 °C (100 m)$^{-1}$, which has been tested amongst other
options and deemed to identify TIs most accurately, prevents this.

Earth System Discussions
Open Access Science
Data

We include two additional criteria when identifying TI layers. First, we only identify sections of the profile in which dT/dz stays above the threshold of 0.65 °C (100 m)$^{-1}$ for at least 25 m as TIs, to avoid including measurement artifacts

or highly-localized temperature variability. Second, if dT/dz goes below the threshold for less than 100 m between two TI layers, then these layers are combined into a single TI layer for the current dataset (Kahl, 1990; Gilson et al., 2018).

Once we have identified all TIs within a profile, we determine the depth of each TI as the vertical distance between the TI bottom and top, and the intensity of the TI as the difference between the temperatures at the TI bottom and top

(Gilson et al., 2018). Figure 2 shows an example of our TI identification method, as well as the depth and intensity of the TIs identified.

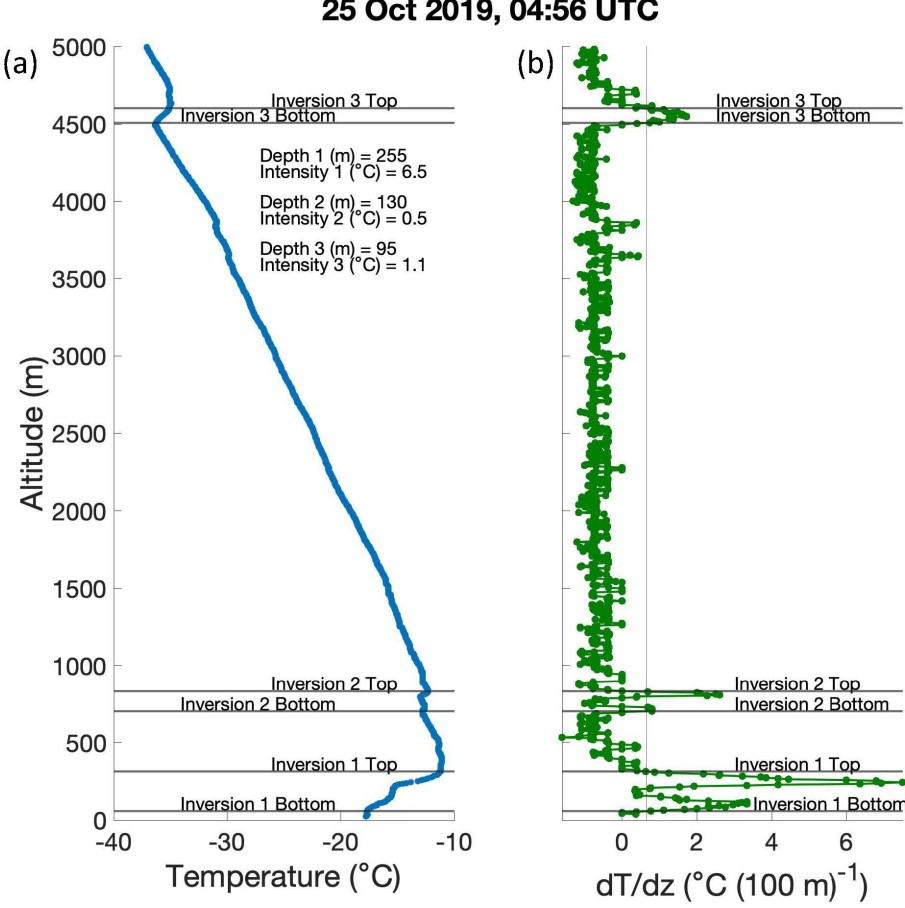

**Figure 2.** Example of temperature inversion identification using radiosonde profile at 04:56 UTC on 25 October 2019. Horizontal black lines on the (a) temperature profile and (b) dT/dz profile indicate the bottom and top of each TI.

Vertical black line on the dT/dz profile indicates the threshold of 0.65 °C (100 m)$^{-1}$. The depth and intensity of each inversion are written on the temperature profile plot.





In the lower atmospheric properties dataset accompanying this paper, the metrics for all TIs found in each radiosonde profile are included in the variables called 'inv_alt' (altitude of the bottom of the TI), 'inv_t' (temperature at the bottom of the TI), 'inv_dt' (TI intensity), and 'inv_dz' (TI depth). These variables are provided as multidimensional

matrices, so that information about all inversions in a given profile, for all profiles, are provided in one variable, with the maximum number of TIs in any one profile being nine.

Additional temperature variables included in the lower atmospheric properties dataset described by this paper are temperature at 2 m and 10 m from the met tower ('t_2m' and 't_10m' respectively), as well as temperature at the top of the ABL from the radiosonde profiles ('t_h'). In addition, pressure at 2 m, 10 m, and ABL top are provided so that

a user can calculate potential temperature at these altitudes ('p_2m', 'p_10m', and 'p_h' respectively).

**3.2 Wind**

The wind-related variables provided in the lower atmospheric properties dataset include low-level jet characteristics as well as zonal and meridional wind speeds from the met tower and at the ABL top.

Using the wind speed profile, we identify an LLJ as a maximum in the wind speed that is at least 2 m s$^{-1}$ greater than

the wind speed minima above and below (Stull, 1988). As described in Tuononen et al. (2015), only situations in which both the wind speed maximum (the LLJ core) and the minima above the core are both below 1500 m are identified as LLJs. Above this altitude, a wind speed maximum is unlikely to be related to surface processes, and more likely to be synoptic in nature. If an LLJ is found, we identify the LLJ core altitude as the altitude of the maximum in the wind speed ('llj_alt'), the LLJ speed as the wind speed at that altitude ('llj_spd') (Jakobson et al., 2013), and the

LLJ direction as the wind direction at that altitude ('llj_dir'). Additionally, we identify the LLJ top as the altitude of the minimum in the wind speed profile above the LLJ. The altitude difference between the LLJ core and top is then the LLJ depth ('llj_dz'), and the difference between the wind speed at the LLJ core and top is then the LLJ strength ('llj_dv') (Jakobson et al. 2013).

In Tuononen et al. (2015) an additional criterion is applied to LLJ identification, in which only a wind speed maximum

that is at least 25% faster than the wind speed at the minimum above is identified as an LLJ. In the lower atmospheric properties dataset, we include an LLJ flag ('llj_flag'), which indicates if the 25% criterion is met (llj_flag = 1) or if it is not (llj_flag = 0). Most instances in which the 25% criterion is not fulfilled are examples in which the wind speed throughout the entire profile is very fast, so the wind speed above the LLJ core decreases by 2 m s$^{-1}$, but not by 25%. We include all LLJs as well as indicate which ones meet this 25% criterion to allow the user to choose which

identification method is relevant to their application of the lower atmospheric properties dataset. Figure 3 shows two examples of our LLJ identification method, one in which llj_flag = 1 and one in which llj_flag = 0, as well as how the depth and strength of the LLJ are calculated. Lopez-Garcia et al. (2022) presents an analysis of MOSAiC LLJ frequency and forcing mechanisms, using only LLJs in which the 25% criterion is met, and thus their analysis is consistent with the LLJ characteristics presented in the lower atmospheric properties dataset when llj_flag = 1.

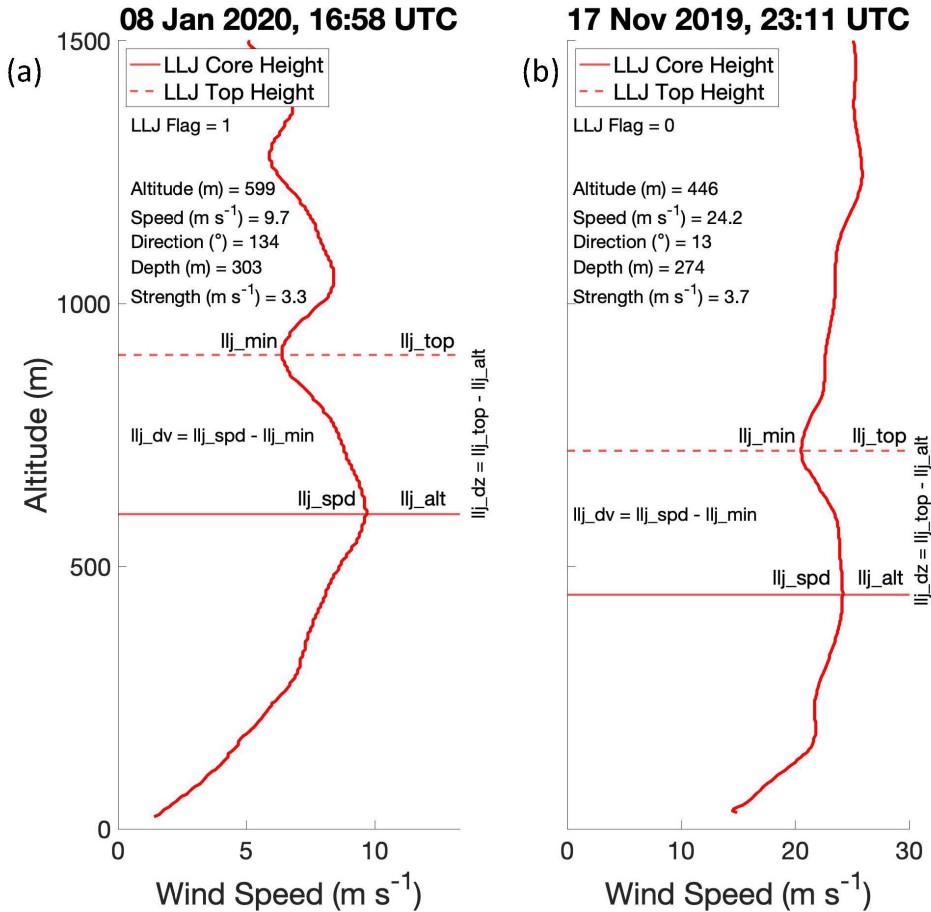


**Figure 3.** Example of low-level jet identification using radiosonde profiles at (a) 16:58 UTC on 8 January 2020 in which llj_flag = 1 and (b) 23:11 UTC on 17 November 2019 in which llj_flag = 0. Horizontal solid red line indicates the altitude of the LLJ core (llj_alt) and horizontal dashed red line indicates the altitude of the top of the LLJ (llj_top). The speed of the LLJ is indicated by llj_spd and the speed at the top of the LLJ is indicated by llj_min. The processes

of calculating LLJ depth (llj_dz) and strength (llj_dv) are shown and all relevant LLJ characteristics are written on both plots.

Additional wind variables included in the lower atmospheric properties dataset accompanying this paper are zonal and meridional wind speed at 2 m and 10 m from the met tower. Zonal wind speed variables are called 'u_2m' and 'u_10m' respectively, and meridional wind speed variables are called 'v_2m' and 'v_10m' respectively. Wind speed

components at the ABL top measured by the radiosonde are also included ('u_h' and 'v_h'). Wind speed is provided in components so the user may calculate total wind speed as well as wind direction, if this is of interest.





### 3.3 Stability

The stability-related variables provided in the lower atmospheric properties dataset include ABL height, the stability
regime both from the met tower and from the lowest portion of the radiosonde measurements, and bulk Richardson
number ($Ri_b$) and change in virtual potential temperature ($d\theta_v$) calculated over three depths: between 2 and 10m,
between the lowest radiosonde measurement and the ABL top, and between 2m and ABL top.

The ABL is the turbulent lowest part of the atmosphere that is directly influenced by the Earth's surface (Stull, 1988;
Marsik et al., 1995). $Ri_b$ is the ratio between buoyantly (from thermals) and mechanically (from wind shear) produced
turbulence (Sivaraman et al., 2013) and can help to identify the top of the ABL under the assumption that turbulence
ceases above of the ABL (Stull, 1988), and thus, $Ri_b$ will exceed a critical value at the top of the ABL (Seibert et al.,
2000). To identify ABL height ('h'), we apply a $Ri_b$-based approach in which the top of the ABL is identified as the
first altitude in which $Ri_b$ exceeds a critical value of 0.5 and remains above the critical value for at least 20 consecutive
meters (Jozef et al., 2022).

$Ri_b$ is calculated using the following equation from Stull (1988):

$$\text{Ri}_b(z) = \frac{\left(\frac{g}{\overline{\theta_v}}\right)\Delta\theta_v\,\Delta z}{\Delta u^2 + \Delta v^2} \tag{1}$$

where $g$ is acceleration due to gravity, $\overline{\theta_v}$ is the mean virtual potential temperature over the altitude range being
considered, $z$ is altitude, $u$ is zonal wind speed, $v$ is meridional wind speed, and $\Delta$ represents the difference over the
altitude range used to calculate $Ri_b$ throughout the profile. $Ri_b$ profiles are created by calculating $Ri_b$ across 30 m
intervals in steps of 5 m and attributing the resulting $Ri_b$ value to the center altitude of $\Delta z$ (i.e., 23-53 m, 28-58 m, 33-
63 m and so on, resulting in a $Ri_b$ profile with values at 38 m, 43 m, 48 m AGL, and so on) rather than using the
ground as the reference level, in order to isolate local likelihood of turbulence rather than that over the full depth from
the surface (Stull, 1988; Georgoulias et al., 2009; Dai et al., 2014). Figure 4 demonstrates an example of how ABL
height is found using this $Ri_b$-based approach. Due to these methods, we cannot identify an ABL height below a
minimum of 38 m (this value may be higher if the bottom altitude of the profile is above 23 m). However, in the case
there is a very shallow ABL due to a surface-based or low-level inversion, this is detected in the first layer of $Ri_b$, and
thus the ABL height is still determined to be shallow.

In addition to the ABL height, we also provide the stability regime (1 = stable boundary layer (SBL), 2 = neutral
boundary layer (NBL), or 3 = convective boundary layer (CBL)) captured by the radiosonde as well as by the met
tower. We provide both, as 45% of the time the stability of the surface layer, recorded by the met tower, was different
than that of the remaining ABL, recorded by the radiosonde. Stability regime from the radiosondes ('s_radiosonde')
and tower ('s_tower') are determined by the following equations (adapted from Liu and Liang (2010) and Jozef et al.
(2022)), which compare $\theta_v$ between the upper and lower bounds of an altitude range spanning the lower atmosphere.





$\quad \theta_{v_{upper}} - \theta_{v_{lower}} < -\delta_s = \text{CBL}$ $\hspace{5cm}$ (2)

$\theta_{v_{upper}} - \theta_{v_{lower}} > +\delta_s = \text{SBL}$ $\hspace{5cm}$ (3)

$-\delta_s \leq \theta_{v_{upper}} - \theta_{v_{lower}} \leq +\delta_s = \text{NBL}$ $\hspace{4cm}$ (4)

$\delta_s = \frac{0.2\ \text{K}}{40\ \text{m}} \Delta z$ $\hspace{6cm}$ (5)

$\Delta z \approx 30$ m (for s_radiosonde)

$\quad \Delta z = 7.69$ m (for s_tower)

Here, $\delta_s$ is a stability threshold that represents the minimum $\theta_v$ increase (decrease) with altitude near the surface necessary for the ABL to qualify as an SBL (CBL). If this minimum is not reached in either direction, the ABL is identified as an NBL (Liu and Liang, 2010). For profiles over ocean/ice, Liu and Liang (2010) define $\delta_s$ to be 0.2 K. Jozef et al. (2022) found that, for shallow Arctic ABLs, comparing the $\theta_v$ change over the lowest 40 m of the profile

$\quad (\theta_{v_{lower}} = \theta_{v_{5m}}$ and $\theta_{v_{upper}} = \theta_{v_{45m}})$ to this stability threshold gives the best estimate of stability regime. Since there are no valid radiosonde observations in any given profile as low as 5 m, and many radiosondes record their lowest good value around 25 m, we adapt the methods presented in Jozef et al. (2022) discussed above to instead compare $\theta_v$ change over the lowest 30 m $(\theta_{v_{lower}} = \theta_{v_i}$ to $\theta_{v_{upper}} = \theta_{v_{i+30m}})$ recorded by the radiosonde for determination of radiosonde stability regime. However, since radiosonde measurements are not taken at the same altitudes in every

$\quad$ profile, we use the altitude range closest to 30 m as possible, but this may vary slightly from profile to profile. Therefore, we use the true $\Delta z$ to calculate a unique $\delta_s$ for each profile, given equation 5. When the top of the ABL is less than 30 m above the lowest radiosonde measurement, we determine stability regime using $\Delta z$ as the distance in meters between the lowest radiosonde measurement and the top of the ABL.

Stability regime from the met tower is found using the altitude range of $\theta_{v_{lower}} = \theta_{v_{2m}}$ to $\theta_{v_{upper}} = \theta_{v_{10m}}$, which

$\quad$ calculates $\delta_s$ using $\Delta z = 7.69$ m, as this is the true distance between met tower measurements at 2 and 10 m, indicated by Table 1.

Additional metrics provided for stability in the lower atmospheric properties dataset are $\text{Ri}_b$ and $d\theta_v$ calculated over different distances in the near-surface layer of the atmosphere. First, $\text{Ri}_b$ and $d\theta_v$ between 2 and 10 m are provided in variables called 'rib_tower' and 'dptv_tower' respectively, calculated using data from the met tower where a positive

$\quad$ value for $d\theta_v$ indicates that virtual potential temperature at 10 m is greater than that at 2 m. Next, $\text{Ri}_b$ and $d\theta_v$ between the bottom of the radiosonde profile and the top of the ABL are provided in variables called 'rib_radiosonde' and 'dptv_radiosonde', respectively. Finally, $\text{Ri}_b$ and $d\theta_v$ between 2 m from the met tower and the top of the ABL from the radiosonde are provided in variables called 'rib_2m_h' and 'dptv_2m_h', respectively.

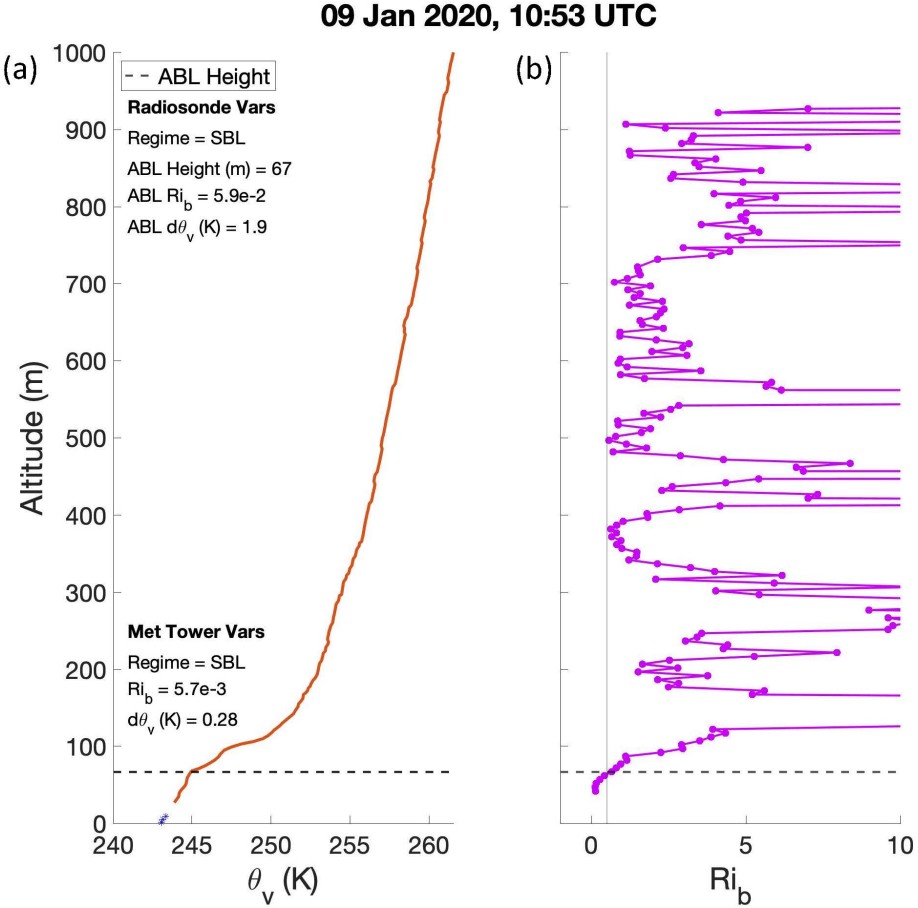

**Figure 4.** Example of atmospheric boundary layer height identification using radiosonde profile at 10:53 UTC on 9 January 2020, where the orange line is radiosonde data and the near-surface blue asterisks are met tower data in (a). Horizontal dashed black lines on the (a) virtual potential temperature profile and (b) $Ri_b$ profile indicate the ABL height, which is also written on the left panel. The stability regime, $Ri_b$ and $d\theta_v$ calculated using both the radiosonde and met tower data are written on the left panel.

### 3.4 Moisture

The moisture-related variables provided in the lower atmospheric properties dataset include the altitude of the lowest cloud base, the percentage of time in the 5 minutes before and after radiosonde launch in which there are clouds (general cloud cover), the percentage of the time in the 5 minutes before and after radiosonde launch in which there are clouds at or below ABL height (ABL cloud cover), and the mixing ratio measured at the met tower and at the ABL top.

Cloud variables are determined using ceilometer data. First, the altitude of the lowest cloud base ('cbh') is determined by calculating the average of all valid observed lowest cloud base heights in the observation interval (approximately





5 minutes before to 5 minutes after radiosonde launch). If there is any period of clear sky during this interval, the clear sky period is excluded from the calculation of the mean.

Next, general cloud cover ('cc') is calculated first by determining if there is any cloud base height detected in the observational interval. Then, we count the number of observations within the observational interval in which there are clouds detected, and divide this by the total number of observations in the interval. Lastly, ABL cloud cover ('cc_h') is estimated by counting the number of observations in the interval in which cloud cover is detected at or below the ABL top, and dividing this by the total number of observations in the interval. Since the ceilometer is located on the

deck of the Polarstern, this method likely misses fog events. For presence of fog, the "present weather" variable in Schmithüsen and Raeke (2021a, b, and c) can be used, though this information on fog is not included in the lower atmospheric properties dataset.

Finally, we provide the mixing ratio at 2 m and 10 m measured by the met tower ('r_2m' and 'r_10m' respectively), and at the top of the ABL measured by the radiosonde ('r_h').

**3.5 Radiation**

The radiation-related variables provided in the lower atmospheric properties dataset include surface upwelling and downwelling broadband longwave and broadband shortwave irradiance, measured by the radiation station located on the MOSAiC floe. For cases in which the sun was below the horizon, the shortwave irradiance recorded may have been a small positive or negative number, due to instrument uncertainty, when the true irradiance is zero. To mitigate

this, we set the average shortwave irradiance over the observational interval to zero if the average solar zenith angle > 93 degrees (the sun was below the horizon and diffuse radiation negligible), or if the average shortwave irradiance is negative.

The up- and downwelling broadband longwave irradiance variables are called 'lwup' and 'lwdn' respectively, and the up- and downwelling broadband shortwave irradiance variables are called 'swup' and 'swdn' respectively. A user may

refer to these values to help calculate total upwelling and downwelling radiation, as well as the surface net radiation.

**4 Summary of the lower atmospheric properties dataset**

Table 3 below summarizes the name of each variable included in the lower atmospheric properties dataset, the quantity it measures (including units), and the platform from which the data came. In addition to the atmospheric properties included in the table, the lower atmospheric properties dataset also includes the latitude ('lat'), longitude ('lon') and

time ('time') of each radiosonde launch, in seconds since Epoch.





**Table 3:** List of variable descriptions, names, units, and the platform from which they were derived that are included in the lower atmospheric properties dataset summarized in this paper. Platform name 'sonde' refers to the radiosondes, 'tower' refers to the 10 m meteorological tower, 'ceil' refers to the ceilometer, and 'radstat' refers to the radiation station.


| | Atmospheric Property | Variable | Units | Platform |
|---|---|---|---|---|
| **Temperature** | temperature at 2 m altitude | t_2m | °C | tower |
| | temperature at 10 m altitude | t_10m | °C | tower |
| | temperature at ABL top | t_h | °C | sonde |
| | pressure at 2 m altitude | p_2m | hPa | tower |
| | pressure at 10 m altitude | p_10m | hPa | tower |
| | pressure at ABL top | p_h | hPa | sonde |
| | lower boundary of each temperature inversion | inv_alt | m | sonde |
| | lower boundary temperature of each temperature inversion | inv_t | °C | sonde |
| | intensity of each temperature inversion | inv_dt | °C | sonde |
| | depth of each temperature inversion | inv_dz | m | sonde |
| **Wind** | zonal wind at 2 m altitude | u_2m | m s$^{-1}$ | tower |
| | zonal wind at 10 m altitude | u_10m | m s$^{-1}$ | tower |
| | zonal wind at ABL top | u_h | m s$^{-1}$ | sonde |
| | meridional wind at 2 m altitude | v_2m | m s$^{-1}$ | tower |
| | meridional wind at 10 m altitude | v_10m | m s$^{-1}$ | tower |
| | meridional wind at ABL top | v_h | m s$^{-1}$ | sonde |
| | low-level jet core altitude | llj_alt | m | sonde |
| | low-level jet core speed | llj_spd | m s$^{-1}$ | sonde |
| | low-level jet core direction | llj_dir | ° | sonde |
| | low-level jet depth | llj_dz | m | sonde |
| | low-level jet strength | llj_dv | m s$^{-1}$ | sonde |
| | low-level jet flag | llj_flag | unitless | sonde |
| **Stability** | $\Delta\theta_v$ between 2 m altitude and 10 m altitude | dptv_tower | K | tower |
| | $\Delta\theta_v$ over the radiosonde data up to ABL top | dptv_sonde | K | sonde |
| | $\Delta\theta_v$ between 2 m altitude and ABL top | dptv_2m_h | K | sonde, tower |
| | ABL top | h | m | sonde |
| | $Ri_b$ between 2 m altitude and 10 m altitude | rib_tower | unitless | tower |
| | $Ri_b$ over the radiosonde data up to ABL top | rib_sonde | unitless | sonde |
| | $Ri_b$ between 2 m altitude and 10 m altitude | rib_2m_h | unitless | sonde, tower |
| | stability regime based on tower data | s_tower | unitless | tower |
| | stability regime based on radiosonde data | s_sonde | unitless | sonde |
| **Moisture** | lowest cloud base altitude | cbh | m | ceil |
| | cloud cover | cc | # out of 1 | ceil |
| | ABL cloud cover | cc_h | # out of 1 | ceil |
| | mixing ratio at 2 m altitude | r_2m | g kg$^{-1}$ | tower |
| | mixing ratio at 10 m altitude | r_10m | g kg$^{-1}$ | tower |
| | mixing ratio at ABL top | r_h | g kg$^{-1}$ | sonde |
| **Radiation** | longwave downwelling radiative flux | lwdn | W m$^{-2}$ | radstat |
| | longwave upwelling radiative flux | lwup | W m$^{-2}$ | radstat |
| | shortwave downwelling radiative flux | swdn | W m$^{-2}$ | radstat |
| | shortwave upwelling radiative flux | swup | W m$^{-2}$ | radstat |



For all variables in the lower atmospheric properties dataset, missing values are given the "_FillValue" and "missing_value" attributes of -9999. When the platform listed in Table 3 is "tower" or "radstat", a missing value means that the tower or radiation measurement was not taken, respectively. When the platform listed in Table 3 is the combined "sonde, tower", a missing value means that the tower measurement needed to determine the quantity was not taken. When the platform listed in Table 3 is "sonde", a missing value indicates that the feature was not present, though the measurement was still taken (e.g., a missing value for llj_alt or inv_alt indicates there was no LLJ or TI present in the observation, respectively). For the "ceil" observations, times when cloud measurements were taken but clouds were not present can be identifiable when there is a missing value for cbh, and cc=0. When there is a missing value for both cbh and cc, then no cloud measurement was taken.

## 5 Data availability

The lower atmospheric properties dataset described in this paper will be available at the PANGAEA Data Publisher once the curation process is complete, but is temporarily available at https://www.pangaea.de/tok/888701783039475ee6c7079e6a380a34fa652fb8 (Jozef et al., 2023) during the review process of this manuscript. Level 2 radiosonde data used to develop the lower atmospheric properties dataset are available at the PANGAEA Data Publisher at https://doi.org/10.1594/PANGAEA.928656 (Maturilli et al., 2021). Near-surface atmospheric data from the meteorological tower and data from the radiation station are available at the National Science Foundation Arctic Data Center (https://doi.org/10.18739/A2PV6B83F; Cox et al., 2023) as described in Cox et al. (submitted). Ceilometer data are available at the Department of Energy Atmospheric Radiation Measurement Data Center at http://dx.doi.org/10.5439/1181954 (ARM user facility, 2019).

## 6 Conclusions

The quantities in the lower atmospheric properties dataset are based on data from 1509 radiosonde profiles, launched between 1 October 2019 and 1 October 2020 at latitudes between 78.36 and 90°N. A wide variety of atmospheric conditions were sampled throughout the MOSAiC year, which will aid interested researchers in understanding the complex interactions between lower atmospheric processes in the central Arctic and their impact on future climate change.

Atmospheric observations from the MOSAiC expedition provide novel insight into the thermodynamic and kinematic processes prevalent in the lower Arctic atmosphere, through the merging of disparate, yet complementary in situ observations. This paper summarizes a dataset that includes information about key atmospheric features observed over the span of an entire year in the central Arctic: the atmospheric boundary layer, temperature inversions, and low-level jets. The lower atmospheric properties dataset also includes information about the state of the near-surface atmosphere, cloud cover, and surface radiation budget. While this paper does not delve into the physical significance of the variables included in the lower atmospheric properties dataset, the authors intend this dataset to be used for a wide variety of applications, including identifying certain times in which features of interest occurred, putting other data into perspective with understanding of the atmospheric state throughout the year, or comparing the characteristics of



different features to each other, with the overall goal of gaining a greater understanding of the atmosphere processes governing the central Arctic and how they may contribute to future climate change.

**Author contributions**

GJ, RR, JC, GdB and BM conceptualized the analysis presented in this paper; SD provided the radiosonde data; CC provided the meteorological tower and radiation data; GJ analyzed the radiosonde, met tower, and radiation data; RR analyzed the ceilometer data; GdB, JC, RR, SD and CC provided feedback on the analysis techniques; GJ wrote the manuscript; RR, GdB, JC, BM, SD and CC reviewed and edited the manuscript.

**Competing interests**

The authors declare that they have no conflict of interest.

**Acknowledgments**

Data used in this paper were produced as part of RV *Polarstern* cruise AWI_PS122_00 and of the international Multidisciplinary drifting Observatory for the Study of the Arctic Climate (MOSAiC) with the tag MOSAiC20192020. We thank all those who contributed to MOSAiC and made this endeavor possible (Nixdorf et al., 2021). Radiosonde
data were obtained through a partnership between the leading Alfred Wegener Institute (AWI), the Atmospheric Radiation Measurement (ARM) User Facility, a US Department of Energy (DOE) facility managed by the Biological and Environmental Research Program, and the German Weather Service (DWD). Meteorological tower data were obtained by the National Oceanographic and Atmospheric Administration (NOAA). Ceilometer data were obtained by the AWI and DOE-ARM User Facility. Radiation data were obtained by the DOE-ARM User Facility. We
appreciate comments provided by an anonymous internal reviewer at NOAA.

**Financial support**

Funding support for this analysis was provided by the National Science Foundation (award OPP 1805569, de Boer, PI), the National Aeronautics and Space Administration (award 80NSSC19M0194), and the German Federal Ministry of Education and Research (award 03F0871A). The meteorological tower and radiation station observations were
supported by the National Science Foundation OPP-1724551, by NOAA's Physical Sciences Laboratory (PSL) (NOAA Cooperative Agreement NA22OAR4320151) and by NOAA's Global Ocean Monitoring and Observing Program (GOMO)/Arctic Research Program (ARP) (FundRef https://doi.org/10.13039/100018302). Additional funding and support were provided by the Department of Atmospheric and Oceanic Sciences at the University of Colorado Boulder, the Cooperative Institute for Research in Environmental Sciences, the National Oceanic and
Atmospheric Administration Physical Sciences Laboratory, and the Alfred Wegener Institute.





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
