# Peer review of "Derivation and compilation of lower atmospheric properties relating to temperature, wind, stability, moisture, and surface radiation budget over the central Arctic sea ice during MOSAiC"

_Earth System Science Data, 2023_

## Author Comment (AC1)

**Response to Referee 1 Comments**

We would like to sincerely thank Referee 1 for taking the time to read our manuscript and provide their helpful comments. These comments have helped to improve the manuscript. Each referee comment is given below in ***bold italics*** followed by our response to the comment. The line numbers provided in our responses refer to line numbers in the revised manuscript.

***This paper provides a good overview of a data set consisting of important properties of the lower atmosphere during the MOSAiC expedition, combining measurements from radiosondes, a surface micro-meteorological tower, solar and IR radiometers, and a laser ceilometer.***

***Synthesis products of this sort provide a valuable framework for the analysis of other measurements such as surface turbulent exchange and the surface energy budget, aerosol properties, boundary layer clouds, gas phase chemistry, etc. For a large, multidisciplinary project, such as MOSAiC, it is particularly useful for the many different studies that may require one or more the properties documented in this data set to all use the same data or definitions and avoid the potential inconsistencies that would result from multiple groups all calculating various parameters independently. The authors thus provide a valuable service to the MOSAiC community with this data set.***

***The paper is well written, and clearly documents the procedures used to define the various quantities, and the quality control applied. I am happy to recommend publication with only minor editorial revisions.***

Thank you for your positive review of our paper. Below we address each of your comments, and explain how and where changes have been made to the manuscript.

***Detailed comments:***

***L32 – recent estimates of the rate of Arctic warming are even higher than this, up to 4 times global mean (Rantanen, M.; Karpechko, A.Y.; Lipponen, A.; Nordling, K.; Hyvärinen, O.; Ruosteenoja, K.; Vihma, T.; Laaksonen, A. The Arctic has warmed nearly four times faster than the globe since 1979. Commun. Earth Environ. 2022, 3, 168)***

Thank you for noticing this error. We were aware of the Rantanen et al. (2022) publication, but had not updated this manuscript to reflect these new findings. We have now replaced the Overland et al. (2019) reference with Rantanen et al. (2022), and now indicate that:

"The Arctic is warming about four times faster than the rest of the planet (Rantanen et al., 2022), a phenomenon called Arctic amplification (Serreze and Francis, 2006; Serreze and Barry, 2011)…" (line 32).

***L69: "in hopes that the dataset will be useful…" – perhaps "in the expectation that…" would be more appropriate phrasing***

We have revised the sentence to say:

"… the goal is simply to explain the instrumentation (Sect. 2) and methods (Sect. 3) used to develop the accompanying lower atmospheric properties dataset, with the expectation that the dataset will be useful to a wide variety of other projects." (line 67)

***L110: re: flow distortion around ships, Achtert et al. (2015) includes CFD model estimates of the impact of flow distortion on wind profiles immediately above a research vessel. Berry et al (2001), while a pioneering study, focus on the, more extreme, impact on wind measurements on the ship itelf.***

*Achtert, P., I. M. Brooks, B. J. Brooks, B. I. Moat, J. Prytherch, P. O. G. Persson, and M. Tjernström. 2015: Measurement of wind profiles over the Arctic Ocean from ship-borne Doppler lidar. Atmos. Meas. Tech. 8, 4993-5007, doi:10.5194/amt-8-4993-2015*

The Achtert et al., (2015) does better support our statement, and thus, we have changed the reference, and the sentence now reads:

"This helps in also removing faulty wind measurements that occur as a result of flow distortion around the ship (Achtert et al., 2015)…" (line 109).

*L176, delete ", for which an example was given in Sect. 2.2", it's unnecessary repetition.*

This has been removed.

*L188-190: delete ", averaged between 5 minutes before and 5 minutes after the time of launch, following the same time interval format as the met tower data, for which an example was given in Sect. 2.2" – again, unnecessary – you already stated that this was done in "the same manner as for the met tower", further repetition of details isn't needed.*

The first half of the sentence in question ("Variables from the radiation station provided at the time of radiosonde launch were determined using 1 minute data (these 1 minute data are determined in the same manner as the met tower data)") in the original manuscript was intended to indicate that the radiation data in Cox et al. (2023a) was given with 1 min resolution, which was reported as the average of the observations between the minute reported, and the following minute, as with the met data. Thus, we did not already state that the radiation data are averaged over the same +/- 5 min range as the met data. We have revised this sentence to be more clear and concise:

"As the met tower data, radiation station data were provided in 1 minute intervals in Cox et al. (2023a), and were averaged in the same manner as the met tower and ceilometer data to report values at the time of radiosonde launch in the current dataset." (line 186).

*L216: on the issue of classifying inversion layers as distinct or a single inversion, Tjernstrom and Graverson (2009) is also relevant (https://doi.org/10.1002/qj.380)*

This reference has been added:

"Second, if dT/dz goes below the threshold for less than 100 m between two TI layers, then these layers are combined into a single TI layer for the current dataset (Kahl, 1990; Tjernström and Graversen, 2009; Gilson et al., 2018)." (line 213)

*Section 3.1 – The discussion of identifying temperature inversions is fine, and the inversions are of importance in their own right. It is perhaps worth noting, however, that from the perspective of stability (and linking to the Richardson number as an indicator of stability), it is not the presence of an inversion - a positive temperature gradient (regardless of threshold used in classifying it as such) – but the gradient relative the adiabatic lapse rate that is important…or the gradient in (virtual/equivalent) potential temperature.*

This is a good point – we don't want a user to be confused about the utility of the temperature inversion metrics for applications related to stability. We have therefore added the following:

"Note that while the presence and strength of temperature inversions may be relevant for some applications related to static stability, a user is encouraged to utilize metrics provided in Sect. 3.3, or calculate the potential temperature gradient for a case of interest, for a better description of stability." (line 229)

*L270: "Wind speed is provided in components so the user may calculate total wind speed as well as wind direction" – this is a bit arbitrary. If the user is interested simply in the wind (speed and direction, or its components) profile (values at specific heights and times) then it makes no difference whether you provide the components or speed and direction – they have the same storage requirements, and it's easy to convert either to the other. If any averaging is required (in altitude or time) then it is much easier to work with components – avoiding the problem of averaging direction across the 0/360 wrap.*

We have revised our justification of including the 2 m , 10 m, and ABL height wind in components by saying:

"Wind is provided in components for ease of calculating a gradient, or temporal or spatial average of wind direction. Total wind speed and wind direction can be calculated from the components, if this is of interest." (line 271)

*L279: "Rib is the ratio between buoyantly (from thermals) and mechanically (from wind shear) produced turbulence" – this is a little misleading. Rib is the ratio between buoyant and mechanical forcing, rather than turbulent production…for stable conditions there is no buoyant production, there is still a (negative) forcing. I suppose one could argue that this is a negative production of turbulence, but the word 'production' implies a positive value.*

We agree that the way this was previously written is misleading. we have revised the sentence to read:

"$Ri_b$ is the ratio between buoyant and mechanical turbulent forcings…" (line 280)

*L324: "Stability regime from the…" -> "The stability regime from the…"*

This has been fixed (line 323).

---

## Author Comment (AC2)

**Response to Referee 2 Comments**

We would like to sincerely thank Referee 2 for taking the time to read our manuscript and provide their helpful comments. These comments have helped to improve the manuscript. Each referee comment is given below in ***bold italics*** followed by our response to the comment. The line numbers provided in our responses refer to line numbers in the revised manuscript.

***This manuscript nicely introduces and describes an aggregated dataset produced from some of the meteorological observations at the year-long MOSAiC drifting experiment. Thermodynamic measurements from frequent radiosondes and from a meteorological tower installed on the sea ice were combined with ceilometer retrievals and surface broadband radiation. These quantities were further used to calculate the bulk Richardson number, atmospheric stability, to locate temperature inversions, and to quantify the presence of low-level jets.***

***The combination of key measured variables together with derived quantities provides a solid tool, useful to characterize the Arctic atmospheric boundary layer over RV Polarstern during the whole MOSAiC expedition, and provides a valuable common ground that reduces the risk of inconsistencies.***

***The manuscript is well written and describes in detail the methods that were used to process the data, providing also useful examples. I recommend it for publication with minor revisions.***

Thank you for your positive review of our paper. Below we address each of your comments, and explain how and where changes have been made to the manuscript.

***Specific comments:***

***60 – I suggest changing to something like "... including the atmospheric boundary layer (ABL) height and stability ..." as I find that the original formulation is a bit vague, especially since other ABL features are listed afterward.***

We have revised this sentence to say:

"… including the atmospheric boundary layer (ABL) height and stability, temperature inversion (TI) and low-level jet (LLJ) characteristics, near-surface meteorological state, cloud cover, and surface radiation budget over the span of an entire year in the Arctic." (line 60)

***Table 1 - The True "10m" height of the wind seemingly has a typo in the ranges (9.9 - 1.1m) since the central value is 10.34m.***

Thank you for noticing this typo. We have fixed this, and also adjusted other values in the table to be consistent with the final values given in Cox et al. (2023b).

***143-146 - It seems that the temperature and humidity setup listed in Table 2 do not match the text in these lines. The instruments in the text (HMT330 and PTU300) differ slightly from the table (HMT337 and PTU307). Additionally, the 2 m temperature is associated with the PTU in the text and with the HMT in the table.***

Thank you for noticing these inconsistencies. The correct instruments are the HMT337 (all 10 m measurements) and PTU307 (all 2 m measurements). This has been fixed.

***270 - I tend to think that the final user could benefit more from the speed and direction rather than the u and v components, especially in the case of quicklooks to be compared with the LLJ parameters, which have a speed and direction format (Table 3). However, I also see the value of u and v for other purposes.***

The authors believe it is preferrable to keep the u and v components, and have added some more text explaining the reasoning:

"Wind is provided in components for ease of calculating a gradient, or temporal or spatial average of wind direction. Total wind speed and wind direction can be calculated from the components, if this is of interest." (line 271)

**295 - radiosonde profile**

We have added 'radiosonde' to the sentence:

"…if the bottom altitude of the radiosonde profile is above 23 m…" (line 295)